# Noise-tolerant single photon sensitive three-dimensional imager

Patrick Rehain[1,2], Yong Meng Sua[1,2 ✉], Shenyu Zhu[1,2], Ivan Dickson[1,2], Bharathwaj Muthuswamy[1,2], Jeevanandha Ramanathan[1,2], Amin Shahverdi[1,2] & Yu-Ping Huang[1,2 ✉]

Active imagers capable of reconstructing 3-dimensional (3D) scenes in the presence of strong background noise are highly desirable for many sensing and imaging applications. A key to this capability is the time-resolving photon detection that distinguishes true signal photons from the noise. To this end, quantum parametric mode sorting (QPMS) can achieve signal to noise exceeding by far what is possible with typical linear optics filters, with outstanding performance in isolating temporally and spectrally overlapping noise. Here, we report a QPMS-based 3D imager with exceptional detection sensitivity and noise tolerance. With only 0.0006 detected signal photons per pulse, we reliably reconstruct the 3D profile of an obscured scene, despite 34-fold spectral-temporally overlapping noise photons, within the 6 ps detection window (amounting to 113,000 times noise per 20 ns detection period). Our results highlight a viable approach to suppress background noise and measurement errors of single photon imager operation in high-noise environments.

[1] Department of Physics, Stevens Institute of Technology, 1 Castle Point Terrace, Hoboken, NJ 07030, USA. [2] Center for Quantum Science and Engineering, Stevens Institute of Technology, 1 Castle Point Terrace, Hoboken, NJ 07030, USA. ✉email: ysua@stevens.edu; Yuping.Huang@stevens.edu

Three-dimensional (3D) imaging technology has found applications across diverse disciplines, including machine-vision and ranging[1,2], terrestrial mapping[3], remote sensing, and environmental monitoring[4–6]. Active 3D imaging, which captures spatial and temporal information by detecting the reflected probe signal from the scene of interest, is becoming an important tool for extending human's visual perspectives. For such, the detector characteristics of a 3D imaging system, like the operation mode, detection sensitivity, timing resolution, and dynamic range, are critical to the scene reconstruction[1]. The latest single photon detection techniques[7] can faithfully detect light in its smallest quanta (i.e., single photons), enabling active imaging upon very low flux of backscattered photons[8], extending its range to tens of kilometer ranges[9]. Combining single photon detection with computational techniques—such as image reconstruction algorithms[10–13], signal processing[8,14,15], and artificial intelligence assisted imaging[16,17]—has provided imaging capabilities under extreme conditions. For example, leveraging the spatial correlation of the target and the physics of low-flux measurement, first-photon imaging was developed to realize 3D imaging from only one detected photon per pixel, even with the presence of ambient noise photons[15]. More recently, low-flux 3D imaging beyond the line-of-sight has been demonstrated[18,19], where the need for computationally intensive reconstruction algorithms has been significantly relieved by the use of light-cone transform enabled confocal scanning technique[20].

Under practical circumstances, the signal photons usually return along with strong background noise spanning the same spectral and temporal domains, making them indistinguishable to the detector[14,21]. Conventional approaches of isolating signal from noise, including those by matched time-frequency filters, are intrinsically limited by the trade-off between signal detection efficiency and noise rejection[22,23]. This limit applies to any linear-optical filtering approach, including those widely employed by computational enhanced imaging techniques[21,24]. Recognizing this, quantum parametric mode sorting (QPMS) has been proposed for mode-selective quantum frequency conversion[22], following the pioneering studies of quantum pulse gating[25] and quantum optical arbitrary waveform generation and measurement[26]. It is implemented by driving the conversion with pump pulses whose spectral width is comparable to its phase matching bandwidth, i.e., at the edge of phase matching. Under this condition, only signal photons in a single spatiotemporal mode, whose profile can be flexibly tailored by shaping the pump pulses, can be converted efficiently. Undesirable photons in other modes, even if they spectrally and temporally overlap with the signal, are converted with much less efficiency[27,28]. This exotic mode selectivity thus realizes superior nonlinear-optical filtering, which was demonstrated to achieve detection signal to noise more than 40 dB over a linear-optical filtering and detection system, and beat the theoretical limit of an ideal matched filter by 11 dB[22,29].

Active imaging also faces challenges caused by photons backscattered before the scenes of interests. As such, the confocal non-line-of-sight and first-photon imaging techniques would struggle to image a fog-besieged or highly-obscured target. This is because a free running single photon detector is likely to be saturated by photons scattered from the thick fog or the obscurant[20], blinding it to the photons carrying information about the target. Even though estimation algorithms were recently demonstrated to improve the accuracy in arrival-time and reflectivity estimations[14,24,30], those post-processing methods are computationally expensive and incapable of eliminating the fundamental uncertainties arising from Poisson (photon number) noise or distortions induced by the pileup effect.

In this work, we address those limitations and extend active 3D imaging to robust operations in photon-starved and noise-polluted environments through the use of QPMS[22,29]. It harvests the maximum noise rejection by QPMS and efficient photon detection by a silicon avalanche photodiode (Si-APD). The signal to noise advantage is 36 dB over direct detection using an InGaAs single photon detector with a 1 ns gated detection window[29], or 7 dB above the theoretical limit of ideal linear-optical matched filters[22]. This allows active 3D imaging in a scenario where the background is orders of magnitude stronger than the backscattered signal, with 34 times more spectral-temporally overlapping noise than signal photons. In addition, we highlight the extreme ranging resolution of our imaging technique by imaging through a highly reflective obscurant without being impeded by the pileup distortions, dead time, and count rate saturation issues that plague many other single photon imagers.

## Results

**High resolution single photon sensitive 3D imaging.** Our 3D imager utilizes an upconversion single photon detector (USPD) capable of QPMS due to carefully selected pump and probe pulses. The setup is shown in Fig. 1 (see Methods section for details). The pump and probe pulses are carved from a 50 MHz femtosecond mode-locked laser (MLL) using separate sets of cascaded spectral filters. The probe pulses are sent out to raster scan the target scene via an optical transceiver and programmable scanning MEMS mirror. Meanwhile, the pump pulses are sent to a programmable optical delay line (ODL). They are then combined with the backscattered probe signal and sent to the USPD. The internal conversion efficiency of the waveguide is 121% $W^{-1}$ $cm^{-2}$, and the total detection efficiency of the USPD is 3.6% with total dark count rate of 250 Hz (as compared with about 1000 Hz for a typical 1-ns gated InGaAs APD)[29]. The dark counts of USPD are primarily attributed to Raman noise photons generated in the upconversion waveguide[22] as the Si-APDs 50 Hz dark count level is negligible. Note that Raman noise photons are maximally filtered out by QPMS due to the mode selectivity of our imager[29,31]. The robust all-fiber design of our imager reduces the number and footprint of required optical components while simplifying the optical alignment procedure. Finally, a field-programmable-gate-array (FPGA) is employed as the central processor for controlling the MEMS mirror and ODL, and to collect data from the USPD.

To reconstruct the 3D image of the target scene, we measure the time-resolving upconversion signal to retrieve the time-of-flight (ToF) information of the backscattered signal while raster scanning the probe beam across the scene. We use a pixelwise maximum-likelihood value (MLV) approach where a time-resolving measurement is performed for each pixel by counting the number of upconverted photons as a function of the temporal delay between the synchronous probe and pump pulses. The temporal delay is scanned using the programmable ODL where backscattered signal photons that are temporally aligned with the pump pulses in the PPLN will be upconverted efficiently and detected by the Si-APD. In contrast, photons distributed in orthogonal time-frequency modes will be upconverted with very low efficiency, even when they temporally overlap with the pump pulses[29,32]. The scanning of the ODL creates a photon-counting histogram for the ODL points, and the MLV estimate is the point with the most detections. For post-processing, a standard MATLAB median filter is applied to smooth the reconstructed image. The filter converts every pixel value to the median of a $3 \times 3$ region consisting of the pixel and its eight nearest neighbors. This commonly used image processing tool is preferred over mean filtering because it is robust against bias from outlier values. An example of a complete time-resolved (longitudinal) upconversion signal from two different transverse locations is shown in

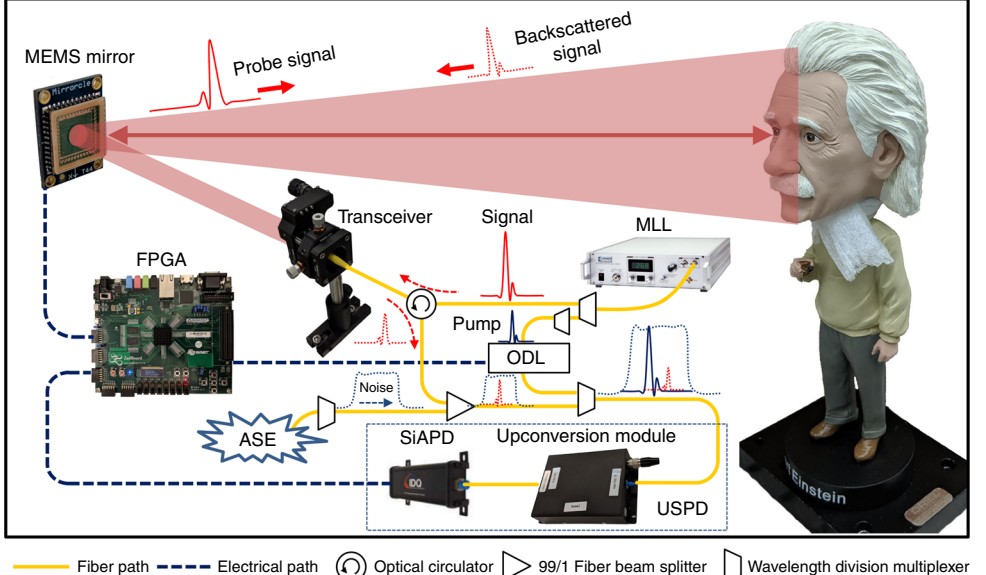

**Fig. 1 Experimental setup of noise tolerant 3D single photon imaging.** MLL, mode-locked fiber laser (Repetition rate = 50 MHz, Center wavelength = 1560 nm); MEMS, micro-electro-mechanical systems; ODL, optical delay line; ASE, amplified spontaneous emission (1520–1570 nm); USPD, upconversion single photon detector; Si-APD, silicon avalanche photodiode; FPGA, field-programmable-gate-array. Pump (1565.5 nm) and probe (1554.1 nm) pulses are carved out from the MLL using off-the-shelf 200 GHz telecom wavelengths dense-wavelength-division-multiplexing (DWDM) filters. The FPGA is the central processor for controlling the programmable MEMS mirror and programmable ODL, and acquiring the data from the USPD. The noise, which is covering the identical spectral-temporal region as probe signal, is carved out from an ASE source with a DWDM filter.

Fig. 2a. Note that the time-resolving upconversion photon counting reflects the intensity-correlation of pump and backscattered probe pulses, where the measured width is about 9 ps (versus ~500 ps for typical InGaAs APDs). This shows the ultrahigh timing resolution of our system upon a single-detection event, which is defined by the phase matching of the upconversion and the full-width-half-maximum (FWHM) of optical pulses. It represents almost one order of magnitude improvement over the resolution achievable with Si-APDs whose timing jitter is typically capped at 50 ps. Sub-picosecond timing resolution is achievable with shorter optical pulses that can be created using spectrally wider filters and waveguides with broader phase matching[33].

Our system can achieve even higher ranging resolution by using the ODL to acquire the full upconversion histogram. We experimentally determine that a peak of 150 detections (equivalent to 0.006 photon detections per pulse over $500\,\mu s$ dwell time) is sufficient to reach the minimum standard deviation of 0.5 ps for a single time-resolving measurement (see Supplementary Note 5). Reduced standard deviation in temporal measurement is translated into improvement in the ranging resolution[15] as depicted in reconstructed 3D image (Fig. 2d).

To assess the depth resolving capability of our imager, we carry out 3D imaging with 2400 pixels on a machined depth resolution chart placed 1.5 m from the transceiver. The chart is a 50 mm × 70 mm aluminum block with 20 depth varying circles of 7 mm diameter (see Fig. 2c). The bottom row contains four reference circles of 1 mm depth. The depth of the remaining circles ranges from 1 to 2.5 mm in steps of 0.1 mm. The reconstructed 3D image is shown in Fig. 2d, where the ODL was scanned in 1 ps steps over a range of 30 ps. The dwell time for each pixel is 1 ms per ODL sample, rendering a total data acquisition time of 72 s. This time can be substantially reduced by using shorter dwell time and fast ODLs, such as those based on solid-state switches. Shown in Fig. 2b, we plot the MLV histograms for all pixels contained within the dotted lines (regions A and B) of Fig. 2c. Each histogram contains two features, with the left-most from

pixels on the surface around the indented circle and the right-most from pixels within the indented circle. The separation between these two features gives a measured ToF difference corresponding to the depth of the circle compared to the surrounding surface. Additionally, the depth profiling accuracy of our 3D imager is determined to be approximately 0.09 mm by measuring it against a calibrated linear translational stage and a certified gauge (see Supplementary Note 4). The transverse spatial resolution of this imager is currently limited by the collimated optical beam diameter (2.2 mm).

**Imaging through an obscuring object**. Our 3D imager is advantageous in discriminating objects of interest in a complex environment with multiple reflecting surfaces owing to the picosecond pump pulses that are physically time-gating the backscattered photons. Therefore, we manage to accept photons from the target while discounting photons arriving at different times, even as close as several picoseconds away. This also effectively removes the distorting pileup effect[14] that can arise from repeated detection of undesired photons within a larger detection window. The performance of our technique shows improvement in terms of sensitivity, timing resolution, accuracy, bias, and dead time[14].

We perform 3D imaging through an obscurant where a ceramic mannequin head is located 2 mm behind an obscuring aluminum wire mesh with 1 mm² holes, as depicted in Fig. 3a, c. Note that the ToF difference between the mesh and the target is well below the 50 ps timing resolution of a typical Si-APD. The highly reflective mesh reduces the backscattered photons from the mannequin by an average of 5 dB while inducing backscattering ahead of the desired target. Nonetheless, our 3D imager is capable of resolving both the mannequin head and the aluminum mesh as shown in Fig. 3b. Additionally, time resolving feature of our imager allows us to isolate mannequin head as depicted in Fig. 3d. Our system distinguishes between the two reflecting surfaces by identifying separate peaks in the upconversion signal. This is

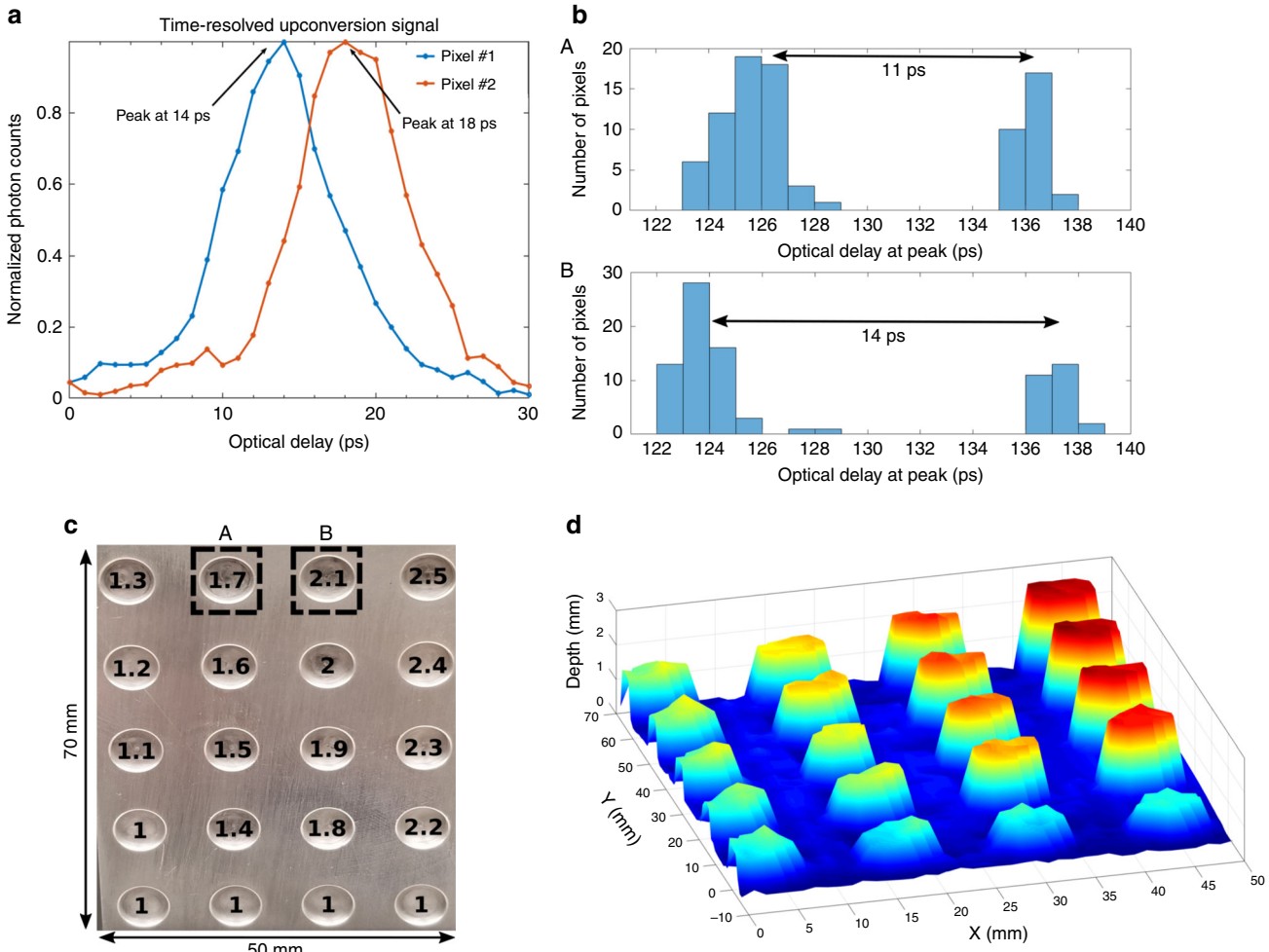

**Fig. 2 High resolution single photon sensitive 3D imaging. a** Time-resolving photon counting from two different longitudinal positions. **b** MLV histogram for two regions (A and B) within the dotted lines in (**c**). **c** 5 cm × 7 cm CNC machined aluminum depth resolution calibration chart. **d** Reconstructed image of the depth resolution chart in (**c**).

demonstrated in Fig. 3e–g which show the time-resolving photon counting measurements for different pixels along the nose of the mannequin head. The left peak represents the backscatter from the mesh and the right peak represents the backscattered signal from the mannequin. The short time-gate created by picosecond pulses enables the immaculate retrieval of 3D information from the mannequin head despite the presence of backscatter from the obscuring material. The optical-gating advantage of our 3D imaging technique can be extended for use in distinguishing objects in a complex environment[19,20] and other applications where precise detection of faint optical pulses is needed[34,35].

**Noise-tolerant imaging.** To demonstrate the noise tolerance of our technique, we perform 3D imaging in a scenario where the backscattered signal photons spectrally and temporally overlap with strong background noise. The noise photons are generated by using a DWDM filter to spectrally carve the amplified spontaneous emission (ASE) from an erbium doped fiber amplifier (EDFA). The filter is similar to the one used to carve the signal from the MLL to ensure that the signal and noise match well in spectrum. Then, we mix the noise with the backscattered signal using a 99:1 fiber beam splitter, as shown in Fig. 1.

We test active imaging under two different levels of background noise, marked by the number of ASE noise photons per period of the signal pulses (20 ns). The results for 350 and 1700

background photons per period are shown in Fig. 4. These noise levels amount to incident signal to background ratios of 1/23,000 and 1/113,000, which is well beyond the capabilities of computational imaging post-processing[36]. For a typical direct detection system using an InGaAs APD with 1 MHz maximum counting rate, 7.5% detection efficiency, and 1 ns time-gating window, those levels correspond to 1.3 and 6.4 mean detected photons per detection window. In both cases the InGaAs detector would be saturated by the noise and therefore unable to detect the low-flux signal. In contrast, the mean photon number detected by QPMS per pulse is only 0.0004 and 0.0016, respectively. Hence, the current QPMS implementation gives an average of about 36 dB advantage in background suppression over direct detection.

This substantial advantage is attributed to the reduced detection mode number, polarization sensitive upconversion, and mode selectivity against spectrally and temporally overlapping noise[17,28,29]. In the case of direct detection, the DWDM filter has the spectral bandwidth of $B = 250$ GHz and detector's gating window $T = 1$ ns, which combined to a total detected time-frequency modes of $\pi BT/2 \approx 392$. For QPMS, the phase matching profile of upconversion waveguide limits the spectral bandwidth to $B = 90$ GHz while the effective detection window of $T = 6$ ps is defined by the FWHM of pump pulse. Evidently, the QPMS reduces the number of detection modes to ≈1, resulting a total noise suppression advantage of 25.9 dB. Theoretically,

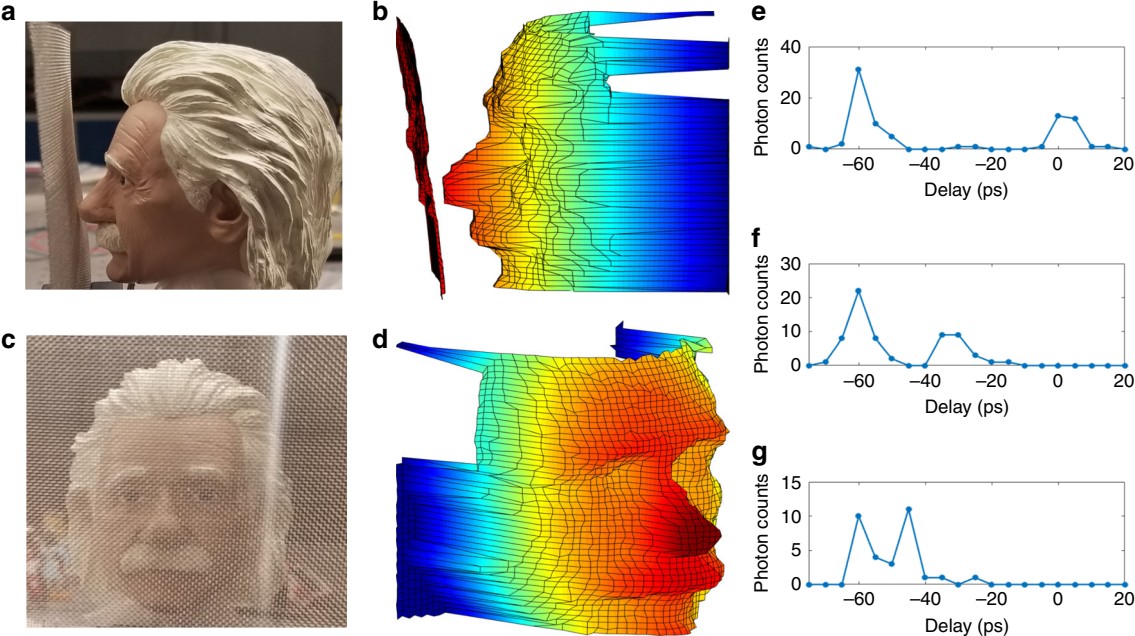

**Fig. 3 Imaging through obscuring object. a** Close-up photo showing the milimetric distance of obscuring aluminum wire mesh (1 mm² opening) and target (head mannequin). **b** Reconstructed 3D image showing the obscurant and target, resembling **a**. **c** Obscuring aluminum grid and target looking at probe signal's propagation direction. **d** Reconstructed 3D image of target isolated from obscurant. **e**–**g** Time-resolving photon counting obtained at different raster scanning position, showing distinguishable backscattering signal peaks from the obscurant and target.

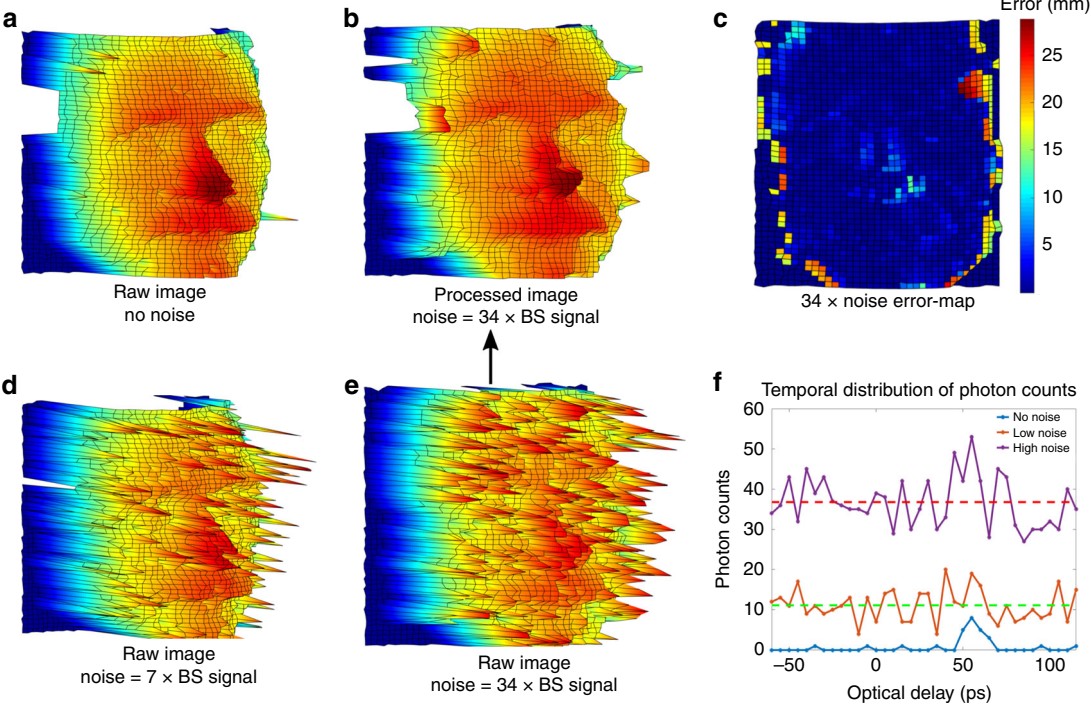

**Fig. 4 Noise-tolerant imaging. a** Noise free raw 3D image of target, acquired with mean photon number of 15 detections per pixel. **b** Reconstructed 3D image acquired with 34 times noise photons after a median filter has been applied. **c** Error map of reconstructed 3D image (**b**), with noise free 3D image (**a**) as ground truth. **d**, **e** The raw 3D images acquired with the presence of 7 and 34 times noise. **f** Time-resolving photon counting at different noise levels, where the dashed lines mark the average number of 7 (green) and 34 (red) times noise photons counted at each ODL point.

similar noise suppression can be achieved by using a combination of ideally matched time-frequency filters. However, it will be difficult to implement in practice for such ultrashort and faint optical pulses. Additionally, upconversion is efficient only along a single polarization, which offers another 3 dB noise suppression by rejecting half of randomly polarized noise photons. Finally, the mode selectivity contributes an additional 7.1 dB to the total 36 dB advantage in background suppression observed, agreeing well with the 7.4 dB mode selectivity predicted by simulation result (see Supplementary Note 3). Even higher selectivity can be

achieved by using a longer PPLN waveguide, tailored phase matching profile or optimally designed pump pulses[27,28,37].

The exceptional noise rejection by QPMS not only defeats the detector saturation, but also helps accurate image reconstruction. This is because high levels of detected noise necessarily leads to image distortion and errors by virtue of its Poissonian variance. Figure 4a, d, e shows how the number of erroneous pixels on the raw 3D images grows quickly with the background level. Figure 4c shows that considerable features on the mannequin head are concealed by the noisy pixels. This highlights the importance of background rejection when the backscattered signal photons are scarce.

Thanks to QPMS' exceptional signal to noise, our 3D imager is able to recover the salient details of the target scene without the use of any post-selective filtering, despite overwhelming noise. In Fig. 4a, b, we compare the reconstructed 3D image without noise and with 1700 background photons per period, acquired with the same signal photon detection rate. Figure 4c shows the depth error (standard deviation) map between these two figures. As seen, the error occurs mainly around high contrast features of the mannequin, e.g., the nose and edge of the face, whose average depth (longitudinal) error is 1 mm. In general, the higher error in those locations is caused by the sharp angle of incidence and thus the dramatically reduced backscattered signal. Consequently, noisy pixels that are not reflecting the true surface morphology are dominant in those areas. Figure 4f gives an example of the time-scanned photon counts acquired by QPMS, where the photon counting peak is still identifiable despite orders of magnitude higher noise. Suppressing noise on the detection end will reduce the number of noisy pixels, allowing for image reconstruction with reduced error. Nevertheless, the errors in the reconstructed 3D image can be further minimized through post-selective filtering that distinguishes between signal and background detections by exploiting the scene's transverse spatial correlation[15].

## Discussion

We demonstrate a noise-tolerant 3D imager that combines the exceptional noise rejection capabilities of QPMS with efficient single photon counting using a Si-APD. The use of QPMS gives a 36 dB advantage in noise rejection over typical direct photon detection with linear-optical filters, or 7.1 dB above the theoretical limit of a matched time-frequency filter. This advantage, made solely on the detection end, allows us to perform 3D imaging with weak returning signal at 0.0006 mean photon detection per pulse despite orders of magnitude stronger background, including the presence of 34-fold spectral-temporally overlapping noise photons. As such, our technique can be particularly useful for extending high-resolution 3D imaging over long distance and strong noise, such as bright solar background[9]. The same advantage may assist deep space communications[38].

Furthermore, with single photon detection of picosecond resolution, our 3D imager achieves 100 μm ranging resolution without any use of computationally intensive image reconstruction algorithms. It thus circumvents the limiting factors of detector time-jittering and pileup distortion, and may find impactful applications in non-line-of-sight and obscured environment imaging. Finally, the present LiDAR technique based on QPMS is applicable to a wide range of wavelengths, including those in the mid-IR regime[31].

## Methods

**Imaging setup**. Two nearly transform-limited, 6 ps pulses at 1554.1 nm (probe) and 1565.5 nm (pump) are carved out from the MLL using a pair of cascaded 200 GHz dense-wavelength-division-multiplexing (DWDM) filters each. We measure the pulse's intensity and phase profile in both spectral and temporal domains using a frequency resolved optical gating (FROG) pulse analyzer,

thereby allowing us to quantify the mode selectivity of upconversion detection[22,29] (See Supplementary Note 3). Collimated signal probe pulses (Gaussian beam diameter: 2.2 mm) at 1554.1 nm are transmitted toward the scene through a transceiver. A fiber circulator separates the outgoing signal pulses and the incoming backscattered photons with a minimum isolation ratio of 55 dB. The transceiver of the imager is based on a simple monostatic coaxial arrangement using off-the-shelf telecom-grade optical components. The back-scattered signal photon will be recombined with pump pulse via another DWDM and subsequently fiber coupled into the mode-selective upconversion detector. Details about the fiber pigtailed detector are explained in Supplementary Note 1. An FPGA is used as central processor for controlling the MEMS mirror and ODL, and acquiring the data from the USPD.

## Data availability

The data that support the findings of this study are available from the corresponding author upon reasonable request.

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

## Author contributions

P.R., Y.M.S., S.Z., I.D., B.M., J.R., A.S., and Y.H. contributed extensively to the work presented in this paper.

## Competing interests

The authors declare no competing interests.
