## [Peer Review File · Nature Communications]

Reviewers' comments:

Reviewer #1 (Remarks to the Author):

In this manuscript, Rehai et al. demonstrate a single-photon level 3D imager based on quantum parametric mode sorting (QPMS). This imager allows for the faithful reconstruction of 3D objects even in the case of strong background noise, which renders standard approaches based on direct detection infeasible. QPMS is used to implement high spectral-temporal noise rejection by effectively implementing a single-mode detection. Every noise photon that is (partially) orthogonal to the detection mode is recorded with diminished probability. Hence, even in the case of noise being orders of magnitudes stronger than the reflected signal light, object reconstruction is possible. In addition, QPMS outperforms linear time-frequency filtering.

The manuscript is well written and the results are impressive. I am wondering, however, whether the paper contains enough genuine novelty to warrant publication in Nature Communications. The authors themselves have demonstrated that QPMS can beat linear filters roughly two years ago. In that sense, the performance of the 3D imager is not surprising. Also, using QPMS as basis for 3D imaging is indeed a brilliant idea, but I want to kindly ask the authors to comment on how this relates to ideas presented by Banaszek and co-workers earlier this year (Banaszek et al., "Approaching the ultimate capacity limit in deep-space optical communication", Proc. SPIE 10910, Free-Space Laser Communications XXXI, 109100A (4 March 2019); <https://doi.org/10.1117/12.2506963>). It seems to me that the idea of using QPMS for time-frequency filtering in LIDAR is very similar to the idea of using quantum pulse gating for noise rejection in deep space communications. I appreciate that this was only presented at a conference and thus is extremely hard to find. Also, I'm only asking for a statement on the similarities between the two approaches to help me make up my mind on the novelty of this manuscript.

There are a few other points I would like to raise and questions I would like to ask:

-- The authors compare their work to direct detection with InGaAs APDs throughout the manuscript, and state that the latter was limited by dark counts, timing resolution, and detector saturation. Can the authors provide any measurement results that support this claim? If not, can the authors comment on the details and assumptions of the underlying model?

-- What is the conversion efficiency of the upconverter used in the experiment?

-- In Figure 4, the authors recover a smoothed image from the raw image by applying a median filter. Maybe they could say a few more words about the parameters of the median filter? Also, it might help the broader readership if the authors explicitly stated that this was a standard method for removing noise from pictures in post processing.

-- Where does the chirp in the pulses in Supplementary Figure 3 originate from? Is it mainly the propagation through the fibres? Or is it from the laser? Or the shaping?

-- Can the authors comment on the possibility to translate this work to other wavelength regimes, e.g. the MIR wavelength region?

-- Can the authors comment on the limits of the timing resolution in their setup? Recently, Donohue et al. demonstrated sub-picosecond timing resolution in single-parameter estimation, which also makes use of projections onto single spectral-temporal modes (Donohue et al., Phys. Rev. Lett. 121, 090501 (2018)).

In conclusion, I think that this manuscript has the potential to be published in Nature Communications. It contains a good idea for a relevant application, the data is sufficient to support the claims of the authors and is well presented, and the work is interesting for a broad readership. If the authors convincingly lay out the novelty contained within this work, I am happy to reconsider the manuscript. At the current stage, I cannot, however, recommend publication in Nature Communications as I am not convinced that the manuscript does contain enough genuine novelty. Still, I want to commend the authors for making reading the manuscript a very joyful experience, and I am sure that they will be able to highlight the novelty more clearly.

Reviewer #2 (Remarks to the Author):

The submitted manuscript introduces and demonstrates a 3D imager concept that uses quantum parametric mode sorting (QPMS) to greatly improve the ratio of detected signal to detected background photons, which I will call detected SBR. This improvement through optical means allows the imager to be effective even when the incident SBR is very low, including settings where the incident background flux would cause prohibitive dead time effects. Since the mitigation of low incident SBR is optical, it does not rely on assumptions about scene structure, as most computational imaging methods for mitigating low incident SBR would do. The avoidance of being limited by dead time effects is well-illustrated by an experiment in which a mannequin head is imaged through a wire mesh (openings about half the spot size diameter, if I have understood correctly) that is very close to the mannequin. Ambient light noise rejection is well-illustrated in a separate experiment.

The authors demonstrate a method for separation of signal and ambient light photons that could have widespread application where it is not precluded by the time needed to scan the setting of a programmable optical delay line (ODL). Though the submitted manuscript depends heavily on its reference [22] (a Scientific Reports paper from the same group), I recommend publication in Nature Communications; the application of QPMS to lidar is a significant achievement beyond what is described in [22]. The submitting group's Frontiers in Optics / Laser Science 2018 abstract "Noise tolerant LIDAR via mode selective up-conversion detection" describes QPMS applied to lidar, but that is merely a short abstract. The biggest weakness of the manuscript is misplaced emphases, some of which are important enough that revision should be considered necessary for acceptance.

In my opinion, the manuscript is fundamentally about achieving "signal" flux measurement with fine time resolution that, because of QPMS, is highly robust to high flux (signal or ambient light) at neighboring time windows. The fine time resolution is illustrated through the experiment of Figure 2; the robustness to high signal flux at nearby earlier time windows is illustrated through the experiment of Figure 3; and the robustness to high ambient light flux at all time windows is illustrated through the experiment of Figure 4.

This basic message of the paper is not clear because of putting "... a few backscattered photons ..." in the opening sentence, and "With only 0.0006 detected signal photons per pulse, we reliably reconstruct the 3D profile of an obscured scene, ..." later in the abstract. The low number of detected signal photons per pulse is almost irrelevant to your manuscript, and it gives the impression that your contributions are about overall photon efficiency in lidar (which it is not). Furthermore, highlighting that you have a low number of detected signal photons per pulse is not consistent with the many mentions of dead time. Low incident flux makes dead time irrelevant, and a reader might naturally multiply 0.0006 times 34 and think that even with signal and ambient light together, the flux is too low for dead time effects to be significant. Furthermore, a reader might naturally look at the factor of 34 and think that this is not radically impressive compared to the results in [Joshua Rapp and Vivek K

Goyal, "A Few Photons Among Many: Unmixing Signal and Noise for Photon-Efficient Active Imaging," IEEE Trans. Computational Imaging, vol. 3, no. 3, pp. 445-459, September 2017] (which, with apologies for advertising my own work, is quite relevant here). Actually, if I am understanding correctly, the factor of 34 that you highlight in the abstract is extremely impressive because it is the ambient light to signal ratio within a 1 ns window. (If you were to compute a signal-to-background ratio over a full 20 ns repetition period, would it be 1 to 680?)

In my opinion, the emphasis of the paper should be that the combination of QPMS and sweeping of a programmable ODL provides capabilities far beyond what is possible with sweeping of the time gating of an ordinary single-photon detector (both because of dead time and the fact that the detected SBR is no higher than the incident SBR). However, taking an entire millisecond for each ODL delay value causes the data collection to be quite slow, and this should be made plain. I feel it takes a bit too much work for the reader to extract this.

Stylistic suggestions and minor corrections:

"maximum-likely-value" is a very uncommon term compared to "maximum likelihood".

I find it a bit suboptimal to use Fig. 2(d) in reference to the to variance of time-of-flight measurements before the paragraph that describes the experiment that yields Figure 2. Fig. 2(c) is used even one paragraph earlier, but this is less jarring because Fig. 2(c) is easier to interpret without understanding the rest of Figure 2.

There is a dimensional mismatch when you write that there is a variance of 0.5 ps. Do you mean to write standard deviation, or is this a variance in ps^2 ?

Is there a reason for the mismatch between "The dwell-time for each pixel is 1 ms per ODL sample" and the 0.5 ms dwell-time in the previous paragraph?

Figure 2(a) is distorted by displaying a non-square object as a square.

-Vivek Goyal

Reviewer #3 (Remarks to the Author):

The manuscript "Noise tolerant singlephoton sensitive 3D imager" from Rehai et al. report on a QPMS based 3D Si-APD imager . Using quantum parametric mode sorting (QPMS), they can suppress the effect of background noise in single photon counting range imaging in photon starved and noise polluted environments.

To realize QPMS, they use a pump and probe setup with a 50 MHz mode locked laser. The method is applied to different scenarios and show remarkable results (for instance, 3d imaging of an artificial had behind an aluminium mesh).

I see no reasons to request any changes and support publication as submitted.

**Reviewer # 1:**

*In this manuscript, Rehai et al. demonstrate a single-photon level 3D imager based on quantum*
*parametric mode sorting (QPMS). This imager allows for the faithful reconstruction of 3D objects even in*
*the case of strong background noise, which renders standard approaches based on direct detection*
*infeasible. QPMS is used to implement high spectral-temporal noise rejection by effectively implementing*
*a single-mode detection. Every noise photon that is (partially) orthogonal to the detection mode is*
*recorded with diminished probability. Hence, even in the case of noise being orders of magnitudes*
*stronger than the reflected signal light, object reconstruction is possible. In addition, QPMS outperforms*
*linear time-frequency filtering.*

*The manuscript is well written and the results are impressive. I am wondering, however, whether the*
*paper contains enough genuine novelty to warrant publication in Nature Communications. The authors*
*themselves have demonstrated that QPMS can beat linear filters roughly two years ago. In that sense, the*
*performance of the 3D imager is not surprising. Also, using QPMS as basis for 3D imaging is indeed a*
*brilliant idea, but I want to kindly ask the authors to comment on how this relates to ideas presented by*
*Banaszek and co-workers earlier this year (Banaszek et al., "Approaching the ultimate capacity limit in*
*deep-space optical communication", Proc. SPIE 10910, Free-Space Laser Communications XXXI,*
*109100A (4 March 2019); <https://doi.org/10.1117/12.2506963>). It seems to me that the idea of using*
*QPMS for time-frequency filtering in LIDAR is very similar to the idea of using quantum pulse gating for*
*noise rejection in deep space communications. I appreciate that this was only presented at a conference*
*and thus is extremely hard to find. Also, I'm only asking for a statement on the similarities between the*
*two approaches to help me make up my mind on the novelty of this manuscript.*

**Reply:** Thanks. We appreciate the findings that “*The manuscript is well written and the results are*
*impressive,*” and that “*using QPMS as basis for 3D imaging is indeed a brilliant idea.*” Indeed, while we
have previously established the principles of QPMS, it is employed here, for the first time, to achieve
unmatched 3D imaging capabilities that herald many practical applications of importance. This could be a
crucial step of transitioning a theory to real-life utilities that can impact a breadth of fields.

The significance and novelty of the present results are also affirmed by Reviewer 2 and 3, who
commented “*The method is applied to different scenarios and show remarkable results,*” and that
“*Though the submitted manuscript depends heavily on its reference [22] (a Scientific Reports paper from*
*the same group), I recommend publication in Nature Communications; the application of QPMS to lidar*
*is a significant achievement beyond what is described in [22].*”

We conceptualized QPMS upon our previous demonstrations of mode-resolving photon counting
(MSPC) [Optics letters 38, 468, Optics Express 22 (23), 27942], which followed the pioneering idea of
quantum pulse gating (QPG) but utilized optical arbitrary waveforms instead of relying on waveguide
engineering. Yet, both QPG and MSPC were targeted at quantum communications and computing, by
discriminatively detecting single photons in a certain time-frequency mode out of several well-defined
modes for information coding. In contrast, QPMS was proposed and demonstrated to address a distinct
challenge facing LiDAR applications: selecting photon signals in a desirable mode mixed with broadband
background noise that resides in numerous, random modes.

Experimental wise, previous QPG and MSPC demonstrations achieved selective photon conversion
over two or several modes created in controlled waveforms. In contrast, we utilized QPMS to select signal
photons over strong noise in lots of random modes (white noise) and scored a huge (~40 dB) advantage in
detection signal to noise ratio.

Thus, while our basic idea shares some similarities with QPG and MSPC, the current work concerns
a different class of applications, and is the first experimental demonstration of QPMS-enabled 3D
imaging, realizing unmatched noise rejection and resolution that promise unprecedented LiDAR and
remote sensing capabilities. The same advantages shall assist deep space communications, as proposed by
Banaszek and co-workers. Our experiment could be viewed a validation of their proposed idea, and we
are interested in further studies in this field.

Changes: In the introduction, we have added: “...following the pioneering studies of quantum pulse
gating [25] and quantum optical arbitrary waveform generation and measurement [26].”

In the summary, we have added “The same advantage may assist deep space communications [38],”
citing the paper by Banaszek and co-workers.

i) *The authors compare their work to direct detection with InGaAs APDs throughout the manuscript, and*
*state that the latter was limited by dark counts, timing resolution, and detector saturation. Can the*
*authors provide any measurement results that support this claim? If not, can the authors comment on the*
*details and assumptions of the underlying model?*

**Reply:** For the 1-ns time gated InGaAs APD, the dark count level was measured to be 1000
counts/second, the timing resolution was measured to be 500 ps and the detector saturates at 10^5
counts/second. In contrast, for the current QPMS detector, the dark count level was measured to be 250
counts/second, the timing resolution to be 9 ps, and the detector saturates at $2 \cdot 10^7$ counts/second.

**Change:** We have now added those numbers in the manuscript; see the track-change version.

ii) -- *What is the conversion efficiency of the upconverter used in the experiment??*

**Reply:** The internal conversion efficiency was measured to be $121 (\%W^{-1} \text{ cm}^{-2})$.

**Change:** While this information was in the section of “Upconversion single photon detector
characterization” in the Supplementary Material, we have now also added this information in the main
text; see the track-change version, page 5.

iii) *In Figure 4, the authors recover a smoothed image from the raw image by applying a median filter.*
*Maybe they could say a few more words about the parameters of the median filter? Also, it might help the*
*broader readership if the authors explicitly stated that this was a standard method for removing noise*
*from pictures in post processing.?*

**Reply:** A median filter was applied to remove erroneous pixels from the reconstructed image by
converting each pixel to the median value of its eight nearest neighbors. These noisy pixels are
singularities that manifest as sharp spikes in the image. The continuous nature of the target scene and the
sparsity of the noisy pixels make this a suitable application for median filtering.

**Changes:** We added the following to the manuscript: “For post-processing, a standard MATLAB median
filter is applied to smooth the reconstructed image by converting every pixel value to the median of its
eight nearest neighbors. This commonly used image processing tool is preferred over mean filtering
because it is robust against bias from outlier values.”

*iv) Where does the chirp in the pulses in Supplementary Figure 3 originate from? Is it mainly the*
*propagation through the fibres? Or is it from the laser? Or the shaping??*

**Reply:** It is due to the flat top in the spectral profile of the commercial dense wavelength division
multiplexers (DWDM-200 GHz) used to carve out probe and pump pulses from a mode-locked fiber laser.

**Changes:** We have added this explanation in the caption of Fig. 3 in Supplementary materials.

*v) Can the authors comment on the possibility to translate this work to other wavelength regimes, e.g. the*
*MIR wavelength region.*

**Reply:** The QPMS LiDAR technique itself can be applied to in principle any wavelength. Limited by its
current implementation using lithium niobite waveguides, the current setting shall be translated to
wavelengths between 0.35 μm to 5 μm ; for a possible realization, see our previous work on “*Direct*
*Generation and Detection of Quantum Correlated Photons with 3.2 μm Wavelength Spacing*” in
*Scientific Reports* 7, 17494.

**Change:** To highlight the wavelength versatility, we have included the following in the Summary:
“*Finally, the present LiDAR technique based on QPMS is applicable to a wide range of wavelengths,*
*including those in the mid-IR regime.*”

*vi) Can the authors comment on the limits of the timing resolution in their setup? Recently, Donohue et al.*
*demonstrated sub-picosecond timing resolution in single-parameter estimation, which also makes use of*
*projections onto single spectral-temporal modes (Donohue et al., Phys. Rev. Lett. 121, 090501 (2018)).*

**Reply:** For our current setup, the timing resolution is limited by full-width-half-maximum (FWHM) of
optical pulses, which can be increased by using a broadband optical filter or waveshaper to carve out
shorter optical pulses, similar to that of the work by Donohue et al. To accommodate those short pulses,
the phase matching bandwidth needs to be increased, too.

**Change:** On page 6, we have added this information; see the track-change version of the revised
manuscript.

**Reviewer # 2:**

*The submitted manuscript introduces and demonstrates a 3D imager concept that uses quantum*
*parametric mode sorting (QPMS) to greatly improve the ratio of detected signal to detected background*
*photons, which I will call detected SBR. This improvement through optical means allows the imager to be*
*effective even when the incident SBR is very low, including settings where the incident background flux*
*would cause prohibitive dead time effects. Since the mitigation of low incident SBR is optical, it does not*
*rely on assumptions about scene structure, as most computational imaging methods for mitigating low*
*incident SBR would do. The avoidance of being limited by dead time effects is well-illustrated by an*
*experiment in which a mannequin head is imaged through a wire mesh (openings about half the spot size*
*diameter, if I have understood correctly) that is very close to the mannequin. Ambient light noise*

*rejection is well-illustrated in a separate experiment.*
*The authors demonstrate a method for separation of signal and ambient light photons that could have*
*widespread application where it is not precluded by the time needed to scan the setting of a*
*programmable optical delay line (ODL). Though the submitted manuscript depends heavily on its*
*reference [22] (a Scientific Reports paper from the same group), I recommend publication in Nature*
*Communications; the application of QPMS to lidar is a significant achievement beyond what is described*
*in [22]. The submitting group's Frontiers in Optics / Laser Science 2018 abstract "Noise tolerant LIDAR*
*via mode selective up-conversion detection" describes QPMS applied to lidar, but that is merely a short*
*abstract. The biggest weakness of the manuscript is misplaced emphases, some of which are important*
*enough that revision should be considered necessary for acceptance.*

**Reply:** Many thanks for the affirmative assessment of novelty and merit, and the recommendation of
publication. Indeed, we hope that this work would inspire and be followed by a multitude of applications
in remote sensing and photon-starved applications.

*In my opinion, the manuscript is fundamentally about achieving "signal" flux measurement with fine time*
*resolution that, because of QPMS, is highly robust to high flux (signal or ambient light) at neighboring*
*time windows. The fine time resolution is illustrated through the experiment of Figure 2; the robustness to*
*high signal flux at nearby earlier time windows is illustrated through the experiment of Figure 3; and the*
*robustness to high ambient light flux at all time windows is illustrated through the experiment of Figure 4.*
*This basic message of the paper is not clear because of putting "... a few backscattered photons ..." in the*
*opening sentence, and "With only 0.0006 detected signal photons per pulse, we reliably reconstruct the*
*3D profile of an obscured scene, ..." later in the abstract. The low number of detected signal photons per*
*pulse is almost irrelevant to your manuscript, and it gives the impression that your contributions are*
*about overall photon efficiency in lidar (which it is not). Furthermore, highlighting that you have a low*
*number of detected signal photons per pulse is not consistent with the many mentions of dead 34 and*
*think that even with signal and ambient light together, the flux is too low for dead time effects to be*
*significant.*

**Reply:** Thanks for this very good suggestion. Yes, the main advantage of this technique is with the noise
rejection and timing resolution, although the imaging capability upon very few returning photons, as
allowed by the very low intrinsic noise of our setup, could be useful for long distance or high loss
applications.

**Change:** We have rephrased the abstract following the suggestion.

*Furthermore, a reader might naturally look at the factor of 34 and think that this is not radically*
*impressive compared to the results in [Joshua Rapp and Vivek K Goyal, "A Few Photons Among Many:*
*Unmixing Signal and Noise for Photon-Efficient Active Imaging," IEEE Trans. Computational Imaging,*
*vol. 3, no. 3, pp. 445-459, September 2017] (which, with apologies for advertising my own work, is quite*
*relevant here). Actually, if I am understanding correctly, the factor of 34 that you highlight in the abstract*
*is extremely impressive because it is the ambient light to signal ratio within a 1 ns window. (If you were*
*to compute a signal-to-background ratio over a full 20 ns repetition period, would it be 1 to 680?)*

**Reply:** Thanks. In fact, the factor of 34 is the ratio of the incident signal photon to **spectral-temporally**
**overlapping** background photons within the 6 ps detection window. This amounts to a factor of 5666 for
a 1 ns window, or 113,320 for a 20 ns window.

**Change:** On page 10, we added: “*These noise levels amount to incident signal to background ratios of*
*1/23,000 and 1/113,000, which is well beyond the capabilities of computational imaging post-processing*
*[36]*”, citing the IEEE paper. We also added this information to the abstract.

*In my opinion, the emphasis of the paper should be that the combination of QPMS and sweeping of a*
*programmable ODL provides capabilities far beyond what is possible with sweeping of the time gating of*
*an ordinary single-photon detector (both because of dead time and the fact that the detected SBR is no*
*higher than the incident SBR). However, taking an entire millisecond for each ODL delay value causes*
*the data collection to be quite slow, and this should be made plain. I feel it takes a bit too much work for*
*the reader to extract this.*

**Reply:** Yes, we agree. For high speed, one will need fast scanning ODL, such as switched solid-state
ODL devices (e.g., <https://agiltron.com/product/rf-microwave-delay-line/>), where the sweeping can
occur at 20 kHz, thereby allowing fast imaging (given enough photon counts).

**Change:** We have added this information on page 7.

*Stylistic suggestions and minor corrections:*

*"maximum-likely-value" is a very uncommon term compared to "maximum likelihood".*

**Change:** Thanks. We have made the change accordingly.

*I find it a bit suboptimal to use Fig. 2(d) in reference to the to variance of time-of-flight measurements*
*before the paragraph that describes the experiment that yields Figure 2. Fig. 2(c) is used even one*
*paragraph earlier, but this is less jarring because Fig. 2(c) is easier to interpret without understanding*
*the rest of Figure 2.*

**Change:** Thanks. We have swapped Fig.2(c) with Fig. 2 (a) to align with the text structure, while
providing side-to-side guide for Fig.2(d) on ranging resolution.

*There is a dimensional mismatch when you write that there is a variance of 0.5 ps. Do you mean to write*
*standard deviation, or is this a variance in ps²?*

**Change:** Thanks for the heads up. We have changed it to the “deviation of 0.5 ps” to avoid confusion.

*Is there a reason for the mismatch between "The dwell-time for each pixel is 1 ms per ODL sample" and*
*the 0.5 ms dwell-time in the previous paragraph?*

**Reply:** 0.5 ms in the dwell-time needed for achieving minimum time deviation of 0.5 ps while the 1 ms
dwell time is used for the acquiring the data for Fig. 2 (d).

**Change:** To avoid confusion, we added the following "...over a range of 30 ps. For Fig. 2 (d), the dwell-
time for each pixel is 1 ms per ODL sample, rendering a total data acquisition time of 72 seconds."

*Figure 2(a) is distorted by displaying a non-square object as a square.*

**Change:** Thanks. We have replotted the figure.

**Reviewer #3:**

*The manuscript "Noise tolerant singlephoton sensitive 3D imager" from Rehai et al. report on a QPMS*
*based 3D Si-APD imager . Using quantum parametric mode sorting (QPMS), they can suppress the effect*
*of background noise in single photon counting range imaging in photon starved and noise polluted*
*environments.*

*To realize QPMS, they use a pump and probe setup with a 50 MHz mode locked laser. The method is*
*applied to different senarios and show remarkeble results (for instance, 3d imaging of an artifical had*
*behind an aluminium mesh).*

*I see no reasons to request any changes and support publication as submitted.*

**Reply:** Thanks a lot. We are encouraged and optimistic to unfold many new opportunities in remote
sensing, imaging, and quantum information.

REVIEWERS' COMMENTS:

Reviewer #1 (Remarks to the Author):

In their reply and resubmitted manuscript, the authors have addressed my concerns re the amount of novelty in their work. I do agree that the novelty here lies in applying the concept of QPMS to 3D imaging. This is indeed an exciting direction and warrants a high profile publication.

In addition, the authors' have also satisfyingly addressed my minor comments and questions.

Thus, I recommend publication of the current manuscript in Nature Communications and congratulate the authors to their very nice work.

Reviewer #2 (Remarks to the Author):

I will refrain from repeating anything from my detailed review of the initial manuscript.

The authors have indeed improved the clarity of the manuscript, especially by making small changes to emphasize the most important novelties of the work.

A few of the changes are not quite correct:

(1) I assume that you have not actually converted "every pixel value to the median of its eight nearest neighbors." Instead, the processing that was used was probably "To smooth the reconstructed image, each pixel value was replaced by the median of a 3×3 neighborhood consisting of the pixel and its eight nearest neighbors."

(2) Instead of replacing "variance" by "deviation," it would be correct to use "standard deviation."

**Reviewer # 2:**

(1) I assume that you have not actually converted "every pixel value to the median of its eight nearest
neighbors." Instead, the processing that was used was probably "To smooth the reconstructed image, each
pixel value was replaced by the median of a 3 \times 3 neighborhood consisting of the pixel and its eight
nearest neighbors."

**Reply:** Thanks. We have made the change the accordingly.

**Change:** On page 6, we added: "For post-processing, a standard MATLAB median filter is applied to
smooth the reconstructed image. The filter converts every pixel value to the median of a 3x3 region
consisting of the pixel and its eight nearest neighbors."

(2) Instead of replacing "variance" by "deviation," it would be correct to use "standard deviation."

**Reply:** Thanks. We have replaced the " deviation " by "standard deviation".

**Change:** on Page 7, "is sufficient to reach the minimum standard deviation of 0.5 ps for a single
time-resolving measurement (see Supplementary Note 5). Reduced standard deviation in temporal
measurement....."